# Use of a filamentous green alga (*Chaetomorpha* sp.) and microsnail (*Stenothyra* sp.) as feed at an early stage of intensive aquaculture promotes growth performance, artificial feed efficiency, and profitability of giant tiger prawn (*Penaeus monodon*)

Isao Tsutsui[1,2]*, Dusit Aue-umneoy[2], Piyarat Pinphoo[3], Worachet Thuamsuwan[3], Kittipong Janeauksorn[3], Grissada Meethong[3], Patcharanut Keattanaworada[3], Jaruwan Songphatkaew[4], Monthon Ganmanee[2], Osamu Abe[1], Kaoru Hamano[5]

1 Fisheries Division, Japan International Research Center for Agricultural Sciences (JIRCAS), Tsukuba, Ibaraki, Japan, 2 Department of Animal Production and Fisheries, Faculty of Agricultural Technology, King Mongkut's Institute of Technology Ladkrabang (KMITL), Bangkok, Thailand, 3 Shrimp Co-culture Research Laboratory (SCORL), King Mongkut's Institute of Technology Ladkrabang (KMITL), Bangkok, Thailand, 4 Department of Fisheries, Ministry of Agriculture and Cooperatives, Bangkok, Thailand, 5 Seikai National Fisheries Research Institute, Nagasaki, Japan

* cong@affrc.go.jp

## Abstract

With the worldwide demand for tropical penaeid prawn increasing in recent decades, more research on shrimp culture methods is needed to enhance efficiency and profitability for shrimp farmers. The objective of this study was to develop a technique to boost the productivity, feed efficiency, and profitability of the giant tiger prawn (*Penaeus monodon*). To accomplish this, a novel culture setup was established in which two benthic organisms, a filamentous green alga (*Chaetomorpha* sp.) and a microsnail (*Stenothyra* sp.), were propagated together with *P. monodon* post-larvae during an early culture stage and then offered to shrimp as supplementary live feeds in intensive aquaculture ponds. For the experiment, shrimp post-larvae (density: approximately 33 individuals m$^{-2}$) were cultured in outdoor concrete ponds (9 × 9 × 1.2 m) under either control (fed only artificial feed, $n = 3$) or experimental (fed artificial feed and benthic organisms, $n = 3$) conditions until they reached marketable size (15 weeks). Apparent green algae consumption was 6.81 kg (8.4% green alga to total feed consumption), whereas microsnail consumption was 1.96 kg (2.4% microsnail to total feed consumption). Compared with the control group of giant tiger prawn, the experimental group showed significantly higher productivity (total number of shrimp produced: 118%; total shrimp production: 133%), feed efficiency (feed conversion ratio of artificial shrimp feed: 89%), and profitability (shrimp sales: 139%; balance between shrimp sales and costs: 146%), while labor and financial costs were kept minimal. These results can be explained by the enhanced growth of shrimp at the early stages of culture. The techniques developed in

**Data Availability Statement:** All relevant data are within the manuscript.

**Funding:** This research was funded by the Japan International Research Center for Agricultural Sciences (JIRCAS)(https://www.jircas.go.jp/en), where Isao Tsutsui (I.T.) and Osamu Abe (O.A.) are affiliates. Also the funders had no role in study design, data collection and analysis, decision to publish, or preparation of the manuscript.

**Competing interests:** The authors have declared that no competing interests exist.

this study will help to advance the efficiency of intensive aquaculture operations for giant tiger prawn and also improve profitability for shrimp farmers.

## Introduction

The global demand for tropical penaeid prawn and their consumption have risen tremendously over the past 30 years, with the total production of aquacultured Penaeidae increasing from 678,000 tons in 1990 to approximately 5.5 million tons in 2017 [1]. White-leg shrimp (*Litopenaeus vannamei*) and giant tiger prawn (*Penaeus monodon*) were the most prominent species of aquacultured Penaeidae, representing 80.9% and 13.4%, respectively, of global Penaeidae production in 2017 [1]. That year, approximately 85% of global Penaeidae production, including 99% of the world's giant tiger prawn production, occurred in Asia [1]. High shrimp production contributes to the economic development of shrimp-producing countries because shrimp is a major export item; for example, the total production of these species for international and domestic consumption generated approximately 1.7 billion USD in Thailand in 2017 [1, 2].

Intensive shrimp culture is the most commonly used method for tropical Penaeidae production, and intensive aquaculture accounted for approximately 98.9% of Penaeidae production in Thailand in 2017 [2]. A crucial issue for farmers of intensive aquaculture is enhancing profitability. Yet decreasing shrimp growth and survival rates have been reported owing to a deterioration in artificial feed quality [3, 4], and the prices of artificial feed have also soared, leading to shrimp feed accounting for approximately 30%–70% of the total aquaculture expenditure in intensive culture systems [5–9]. Consequently, methods involving low production costs, minimal labor, increased productivity, and the effective use of artificial feed are urgently required to improve the profitability of intensive shrimp culture.

To boost productivity and feed efficiency in intensive shrimp culture, the possible use of various benthic organisms as supplementary live feeds has been explored. For example, the filamentous green seaweeds *Ulva* spp., *Chaetomorpha* spp., and Cladophoraceae spp., as well as the red seaweed *Agarophyton* spp. (formerly *Gracilaria* [10]), have been shown to enhance shrimp growth and/or survival [11–17]. However, it is difficult to predict and extrapolate these findings to real-world situations because relatively high amounts of benthos were fed to the shrimp compared with the water volumes or shrimp numbers, and these benthos amounts are not easily attainable in intensive culture ponds. Furthermore, although attention to profitability is necessary to consider the potential implementation of innovative techniques in intensive shrimp ponds, it has rarely been discussed in previous works [10–17]. There are practical issues associated with performing experiments using real intensive shrimp ponds because it is difficult to obtain the necessary numbers of ponds for statistical replication and to scale up the experimental conditions. Therefore, in the present study, a group of outdoor concrete ponds was utilized, providing a comparable experimental scale to the usual culture scale in such terms as numbers of cultured shrimp and water volumes, with statistical replication.

The filamentous green alga, *Chaetomorpha* sp., has very thin, light-to-dark green filaments composed of a series of non-branching cylindrical cells measuring 45–136 μm in length and 27–64 μm in width [18]. The microsnail, *Stenothyra* sp., has a tiny light-to-dark brown shell that is 2–3 mm high in adults [19]. These benthic organisms have no market value as fishery products, but are considered promising candidates as supplementary feeds for tropical Penaeidae. Both benthic species have been shown to promote shrimp growth [16, 20], thrive in a

wide range of shrimp pond conditions, particularly in terms of salinity and temperature [18–20], and grow abundantly in stagnant waters and/or channels in mangrove areas along tropical coasts [20–22].

The aim of this study was to develop a simple and low-cost technique to boost the productivity, feed efficiency, and profitability of the giant tiger prawn. This study's findings advance the implementation of an innovative giant tiger prawn aquaculture system, in which a filamentous green alga (*Chaetomorpha* sp.) and microsnail (*Stenothyra* sp.) are cultured together in an early culture stage and may be freely consumed as supplementary live feeds in intensive aquaculture ponds. Shrimp productivity, feed efficiency, and profitability of giant tiger prawn production were evaluated in experimental outdoor concrete ponds. The benthos amounts collected, propagated, and fed were minimized to the extent that all were consumed within ~1–2 months, to maintain a low level of additional operating efforts. The findings of this study will provide relevant knowledge for the innovative design of commercial-scale earthen intensive farms rearing giant tiger prawn.

## Materials and methods

### Ethical statement

All research activities performed in this study were permitted by the National Research Council of Thailand (NRCT) (Project ID 2011/005). The care and handling of experimental animals were conducted following Japanese and Thai laws. The experiment was performed from the end of August to the beginning of December 2016 at an outdoor experimental facility at the Shrimp Co-culture Research Laboratory (SCORL), King Mongkut's Institute of Technology Ladkrabang (KMITL), Bangkok, Thailand.

### Experimental shrimp, seaweed, and snails

A total of 20,000 individuals from a batch of 12-day-old giant tiger prawn post-larvae (PL) were purchased from a commercial nursery in Chon Buri Province, Thailand. Following transportation to KMITL, the PL were acclimatized to the outdoor water temperature in concrete experimental ponds for approximately 30 min before release.

Benthic species (*Chaetomorpha* sp. and *Stenothyra* sp.) were collected at an abandoned brackish pond in Samut Sakhon Province, Thailand over a period of approximately 3–4 h. After transportation to KMITL, the seaweed and snails were rinsed with fresh water to remove and kill any periphyton.

### Experimental design and culture procedures

In total, six concrete ponds were used, each of which was 9.0 m long × 9.0 m wide × 2.0 m deep. These ponds were arranged in two lines and three rows in the experimental area at KMITL (Fig 1). The water depth was adjusted to approximately 1.2 m in each pond, and aeration was uniformly provided for 24 h using perforated pipes lying on the bottom of each pond and an aeration blower (RT-6037; Ron Tai Electrical Engineering Co., Ltd., Taichung, Taiwan). These ponds were designated as either "artificial feed only" (control) or "artificial feed and benthos" (experimental treatment), with three replicates per treatment. A total of 2,700 PL were released into each pond at an initial density of approximately 33 individuals $m^{-2}$ (approximately 28 individuals $m^{-3}$). Additionally, *Chaetomorpha* sp. and *Stenothyra* sp. with mean fresh weights of 6.81 kg and 1.96 kg, respectively, were released into the experimental treatment ponds. Any benthic organisms other than *Chaetomorpha* sp. and *Stenothyra* sp. were removed by hand, siphon, or filtration during snorkeling observations when possible during

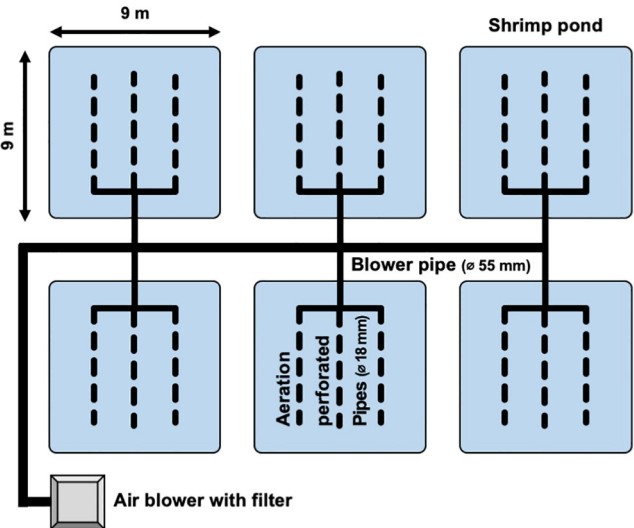

**Fig 1. Schematic diagram of the experimental design.**

the experimental period. The giant tiger prawn in each pond were fed commercial pelleted feed (Profeed; Thai Union Feed Mill Co. Ltd., Samut Sakhon, Thailand) three times per day (9:00, 13:00, and 17:00). The amount of artificial feed provided was determined via the feeding tray method, in which the feed was adjusted to minimize the amount remaining after each feed. This was accomplished by recording the amount of feed remaining on the feeding tray in each pond and using it as an indicator of the appetite of the shrimp. *Chaetomorpha* sp. and *Stenothyra* sp. were freely consumed by the shrimp in the experimental treatment ponds.

## Measurement of monthly shrimp growth and final production

To determine the monthly changes in shrimp weight, the individual wet weights of 20 shrimp that had been randomly collected from each pond were measured using an electric balance (PL 602–5; Mettler-Toledo International Inc., Greifensee, Switzerland) every month (weeks 0, 4, 8, and 12). Prior to measurement, the water on the surface of each shrimp was quickly and carefully wiped using Kimwipes (Kimberly-Clark Corporation, Wisconsin, USA). After measurement, each shrimp was returned to its original pond.

To calculate final production, all of the shrimp were harvested from each pond at the end of the experiment (week 15) and transported to a shrimp broker company in Bangkok. The broker company staff sorted all of the shrimp according to market size categories and determined the shrimp weights and prices in each size category. Productivity, artificial feed efficiency, and profitability were then calculated based on the data obtained from the shrimp broker company.

Total shrimp production ($P_{total}$) was calculated using the formula:

$$P_{total}(\text{kg} \quad \text{WW}) = \sum\nolimits_{k=1}^{n} \text{WCk} \tag{1}$$

where $WC_n$ is the weight of the shrimp in size category $n$ in kg wet weight (kg WW). The wet weight of the 30 shrimp that were randomly sampled from each size category was then measured, and the total number of shrimp produced ($N_{total}$) was calculated using the formula:

$$N_{total}(\text{individuals}) = \sum\nolimits_{k=1}^{n} \frac{WCk}{AWCk} \tag{2}$$

where $AWC_n$ is the average weight of 30 shrimp measured from size category $n$ (kg WW). Survival rate was calculated using the formula:

$$Survival \quad rate(\%) = N_{total}/N_i \times 100 \tag{3}$$

where $N_i$ is the initial number of PL released (2,700 individuals).

The specific growth rate (SGR) was calculated using the formula:

$$SGR(\% \, d^{-1}) = (\ln P_{total} - \ln PLW_i)/D \times 100 \tag{4}$$

where ln is the natural logarithm, $PLW_i$ is the initial total weight (kg WW), and $D$ is the culture duration (days). Finally, the artificial feed conversion ratio (aFCR) was calculated using the shrimp harvest data with the formula:

$$aFCR = F/(P_{total} - PLW_i) \tag{5}$$

where $F$ is the apparent total artificial feed intake.

## Measurement of benthos biomass change and apparent benthos consumption

*Chaetomorpha* sp. and *Stenothyra* sp. were sampled monthly from three replicate quadrats in each experimental treatment pond. The total coverage of the two species was also visually estimated via snorkeling in each of the experimental treatment ponds. The benthic biomass (BB) of each species was then calculated using the formula:

$$BB(\text{kg WW}) = SWQ/QA \times BA \tag{6}$$

where $SWQ$ is the sample weight in each quadrat (kg WW), $QA$ is the quadrat area (m²), and $BA$ is the area of benthos coverage in the pond (m²).

Apparent benthos consumption (ABC) for both the green alga and microsnail was calculated as follows:

$$ABC(\text{kg WW}) = BB_{ast} - BB_{pst} \tag{7}$$

where $BB_{ast}$ is the benthos biomass at any sampling time (kg WW), and $BB_{pst}$ is the benthos biomass at a previous sampling time (kg WW).

## Water quality measurements

The initial aquaculture water was adjusted to a salinity of 25 PSU by adding fresh water to concentrated seawater (approximately 150 PSU) purchased from a reservoir pond for a salt farm in Samut Songkhram Province, Thailand. During the experiment, the salinity in each pond was allowed to fluctuate naturally with rainfall or evaporation and tested every week using a salinity refractometer (Master-S/Mill Alpha; Atago Co. Ltd., Tokyo, Japan). Water temperature and pH were not controlled throughout the experimental period. Water temperature was measured every hour using a temperature logger (UA-002-08; Onset computer Inc., Massachusetts, USA). pH was measured every morning (9:00) using a pH meter (HI98127; Hanna Instruments Inc., Rhode Island, USA). Alkalinity was measured every 2 weeks using the potentiometric titration method [23].

Total ammonia-N and phosphate were analyzed using the Nessler and ascorbic acid methods, respectively, via a portable spectrophotometer (DR 2400; HACH Co. Ltd., Loveland, CO, USA). Nitrite-N and nitrate-N were analyzed using the colorimetric method of the American Public Health Association [23].

### Analysis of the proximate compositions of the feed materials

The approximate compositions of *Chaetomorpha* sp., *Stenothyra* sp., and the artificial shrimp feed were analyzed using the official method of the Association of Official Analytical Chemists [24]. The carbohydrate contents were calculated by subtracting the protein, fat, and ash contents from 100% of the total content.

### Statistical analyses

The productivity, feed efficiency, and profitability of the two experimental groups were compared based on the shrimp harvest and sales data at week 15 using the Student's *t*-test with a significance level of 0.05. The significance of differences in the monthly changes in shrimp growth between the two experimental groups was tested using the Mann–Whitney *U* test, because the data were found to be non-normal through the Shapiro–Wilk test. In this case, the significance level of 0.05 was adjusted via the Bonferroni correction for multiplicity. Differences in monthly changes in SGR between the two experimental groups were tested using the Student's *t*-test, with the significance level of 0.05 again adjusted by the Bonferroni correction for multiplicity. All statistical analyses were performed using the software JMP 10 (SAS Institute, Cary, NC, USA).

## Results

### Shrimp growth and productivity

The individual shrimp weight at week 4 was significantly higher in the experimental treatment ponds (median, 1.12 g; mean, 1.44 g) than that in the control ponds (median, 0.70 g; mean, 0.80 g) (Fig 2). Similarly, the mean monthly SGR for weeks 0–4 was also significantly higher in the experimental group (23.4% day$^{-1}$) than that in the control group (21.4% day$^{-1}$) (Fig 3). After the fourth week of culture, the shrimp in the two groups continued to exhibit significant differences in weight (Fig 2), but no longer differed in their mean SGRs (Fig 3).

The final mean weight of individual shrimp and SGR at week 15 in the experimental treatment ponds were 113.4% and 103.7%, respectively, of the values obtained in the control ponds, representing significant differences (Table 1). Shrimp size variation was greater in the

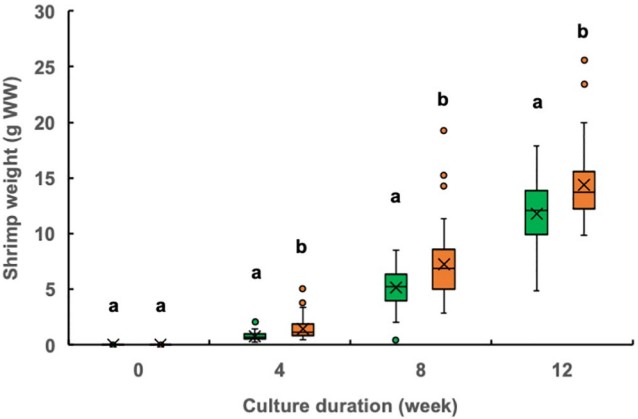

**Fig 2. Box-and-whisker plots of the weights of giant tiger prawn (*Penaeus monodon*) sampled at 0, 4, 8, and 12 weeks under the control (fed only artificial feed) and treatment (fed artificial feed and benthos) conditions.** Green box, control; orange box, treatment; x, mean; horizontal bar within box, median; circle, outlier. Different lowercase letters above the boxes within the same sampling week indicate a significant difference between treatments (Mann–Whitney *U* test, p < 0.05). WW, wet weight.

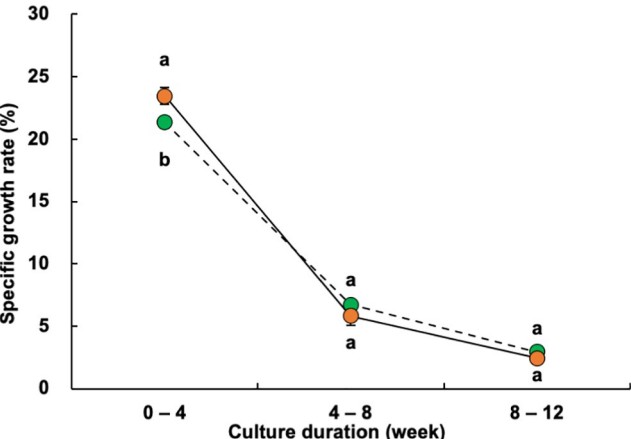

**Fig 3. Changes in the specific growth rates of giant tiger prawn (*Penaeus monodon*) sampled at 0–4, 4–8, and 8–12 weeks under the control (fed only artificial feed) and treatment (fed artificial feed and benthos) conditions.** Green circle, control; orange circle, treatment. Values are means ± standard deviations (SD). Different lowercase letters within the same sampling period indicate a significant difference between treatments (Student's t-test, $p < 0.05$).

experimental treatment ponds, with a higher number of larger shrimp observed (Fig 4). Both the mean total number of shrimp produced and survival rate were 117.5% of the values recorded in the control ponds, while the mean total shrimp production was 133.1% of the values found in the control ponds; between-treatment differences were significant for each parameter (Table 1).

## Feed consumption and efficiency

The crude protein content for 100 g of dried artificial shrimp feed was 39.30 g, while those of fresh *Chaetomorpha* sp. and fresh *Stenothyra* sp. (including the shell) were much lower at 5.16 g and 8.27 g, respectively (Table 2).

The monthly feed intake during the first 4 weeks was higher in the experimental treatment ponds (9.60 kg; artificial shrimp feed:fresh *Chaetomorpha*:fresh *Stenothyra* = 24:55:20) than in the control ponds (1.63 kg) (Table 3). Furthermore, the calculated protein intake in the experimental treatment ponds (1.36 kg; artificial shrimp feed:fresh *Chaetomorpha*:fresh *Stenothyra* = 68:20:12) was approximately 201.2% of that in the control ponds (0.65 kg). All of the *Stenothyra* sp. was consumed and disappeared from the experimental treatment ponds during the first 4 weeks of culture as a result of active shrimp consumption, whereas a small amount of *Chaetomorpha* sp. remained.

In both the control and experimental treatment groups, the consumption of artificial feed sharply increased during weeks 4–8 compared with the previous month, but the feed intake remained higher in the experimental treatment ponds (16.56 kg; artificial shrimp feed:fresh *Chaetomorpha*:fresh *Stenothyra* = 91:9:0) than in the control ponds (10.40 kg). During this period, the calculated protein intake in the experimental treatment ponds was 6.11 kg (artificial shrimp feed:fresh *Chaetomorpha*:fresh *Stenothyra* = 99:1:0), which was approximately 146.8% of that in the control ponds (4.16 kg). *Chaetomorpha* sp. disappeared from the experimental treatment ponds in weeks 4–8 as a result of shrimp consumption, such that after week 8, the shrimp only consumed artificial feed.

The total consumption of artificial shrimp feed throughout the experimental period (15 weeks of culture) in the experimental treatment ponds was 118% of that in the control ponds,

**Table 1. Productivity, feed efficiency, and profitability of giant tiger prawn (*Penaeus monodon*) produced under control (fed only artificial feed) and experimental treatment (fed artificial feed and benthos) conditions for 15 weeks.**

| | Control (*n* = 3) | Experimental treatment (*n* = 3) |
|---|---|---|
| **Growth and productivity** | | |
| Initial mean individual shrimp weight (mg WW) | 2.0 ± 1.0 [a] | 2.0 ± 1.0 [a] |
| Final mean individual shrimp weight (g WW) | 16.0 ± 0.61 [a] | 18.2 ± 1.07 [b] |
| Specific growth rate (% day$^{-1}$) | 8.15 ± 0.05 [a] | 8.41 ± 0.01 [b] |
| Total number of shrimp produced | 2,058 ± 132 [a] | 2,419 ±167 [b] |
| Survival rate (%) | 76.2 ± 4.9 [a] | 89.6 ± 6.2 [b] |
| Total shrimp production (kg WW) | 33.0 ± 1.8 [a] | 43.9 ± 0.5 [b] |
| **Feed intake and efficiency** | | |
| Apparent *Chaetomorpha* intake (kg WW) | N/A | 6.81 ± 1.45 |
| Ratio of *Chaetomorpha* to total feed consumption (%) | N/A | 8.37 ± 1.35 |
| Calculated *Chaetomorpha* protein intake (kg) | N/A | 0.14 ± 0.03 |
| Apparent *Stenothyra* intake (kg WW) | N/A | 1.96 ± 0.05 |
| Ratio of *Stenothyra* to total feed consumption (%) | N/A | 2.43 ± 0.20 |
| Calculated *Stenothyra* protein intake (kg) | N/A | 0.16 ± 0.00 |
| Apparent artificial feed intake (kg WW) | 61.0 ± 3.2 [a] | 72.0 ± 3.8 [b] |
| Calculated artificial feed protein intake (kg) | 24.0 ± 1.3 [a] | 28.3 ± 1.5 [b] |
| Artificial feed conversion ratio | 1.85 ± 0.06 [a] | 1.64 ± 0.10 [b] |
| **Costs and profitability** | | |
| Artificial shrimp feed cost (USD) | 83.55 ± 4.45 [a] | 98.59 ± 5.24 [b] |
| Miscellaneous costs (USD) | N/A | 12.11 ± 0.00 |
| Shrimp sales (USD) | 155.73 ± 10.27 [a] | 215.97 ± 4.37 [b] |
| Balance between shrimp sales and costs (USD) | 72.18 ± 7.55 [a] | 105.27 ± 3.02 [b] |
| Relative profitability to control | 1.00 | 1.46 |

Values shown are the means ± SD for three replicate ponds.

Protein amounts were calculated from the proximate composition data for each kind of feed (Table 2).

Common expenses for the control and experimental treatment, such as water charges, electricity fees, labor costs, and culture materials, have been omitted to more easily compare the profitability results.

Miscellaneous costs include personnel (three people for approximately 4 hours of labor), fuel, and other costs associated with benthos collection.

Different superscript lowercase letters within the same row indicate a significant difference between treatments (Student's t-test, p < 0.05).

Currency exchange rate: 1 USD ≈ 33.02 THB (as of April 2020).

N/A: Not applicable.

which represented a significant difference, and the aFCR of the artificial shrimp feed was significantly lower in the experimental treatment ponds than in the control ponds (Table 1).

## Cost and profitability

Common expenses to both the control and experimental treatments, such as PL costs, water bills, electricity charges, fuel expenses, labor costs, and equipment costs, were omitted from this paper so that profitability could be more readily compared. The artificial shrimp feed cost was significantly higher for the experimental treatment (98.59 USD) than for the control (83.55 USD) (Table 1). Miscellaneous costs, including those associated with the hiring of personnel and fuel for benthos collection, was 12.11 USD (Table 1). The price curve indicated that the individual shrimp price exponentially increased by shrimp weight (Fig 5). The mean shrimp sale price was significantly higher for the experimental treatment group (215.97 USD) than for the control group (155.73 USD) (Table 1). The mean balance between the shrimp sale

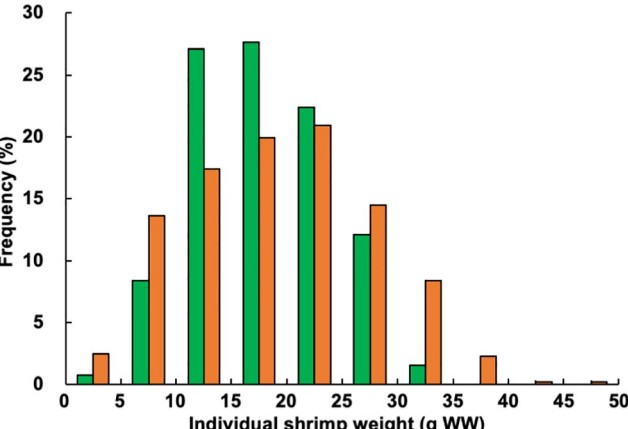

**Fig 4. Size–frequency distributions of giant tiger prawn (*Penaeus monodon*) produced under the control (fed only artificial feed) and treatment (fed artificial feed and benthos) conditions.** Green bar, control; orange bar, treatment. WW, wet weight.

and feed cost in the experimental treatment was 105.27 USD, which was 145.8% of that of the control (72.18 USD), and there was a significant difference between the control and experimental groups (Table 1).

## Water quality

There were no significant differences between the treatment groups in any of the measured water quality parameters (Table 4). However, the mean total ammonia-N tended to be lower in the experimental treatment ponds.

# Discussion

## Shrimp growth and productivity

Various benthic organisms have been used as supplementary live feeds for tropical Penaeidae in previous studies [11–15, 17, 18], and they improved shrimp growth and feed efficiencies. These past studies showed that the final weights and SGRs of tiger prawn in the experimental treatment fed benthic organisms were 115%–157% and 108%–116%, respectively, of the values in the control (only fed artificial feed), and that the final weights and SGRs of white-leg shrimp were 130%–142% and 107%–188%, respectively, of the control values (Table 5). Similarly, in

**Table 2. Proximate compositions of dried artificial shrimp feed, fresh *Chaetomorpha* sp., and the fresh whole body of *Stenothyra* sp. (including shell) per 100 g.**

|  | **Artificial feed** | ***Chaetomorpha* sp.** | ***Stenothyra* sp.** |
|---|---|---|---|
| Moisture | 9.75 ± 0.08 | 83.28 ± 0.68 | 69.31 ± 0.14 |
| Protein (g) | 39.30 ± 0.24 | 5.16 ± 0.04 | 8.27 ± 0.03 |
| Fat (g) | 1.83 ± 0.16 | 0.11 ± 0.01 | 0.30 ± 0.02 |
| Ash (g) | 11.58 ± 0.06 | 3.11 ± 0.02 | 20.66 ± 0.10 |
| Carbohydrate (g) | 47.28 ± 0.25 | 4.24 ± 0.04 | 0.48 ± 0.14 |
| Crude fiber (g) | 3.67 ± 0.12 | 3.88 ± 0.04 | 0.99 ± 0.09 |

Values shown are the means ± SD for three replicate ponds.

Crude fiber is treated as a component of carbohydrate.

**Table 3. Changes in the apparent feed consumption and calculated protein consumption of giant tiger prawn (*Penaeus monodon*) under the control (fed only artificial feed) and experimental treatment (fed artificial feed and benthos) conditions.**

| | Control (*n* = 3) | | Experimental treatment (*n* = 3) | | | | | |
| | Artificial feed | | Artificial feed | | *Chaetomorpha* sp. | | *Stenothyra* sp. | |
| Week | Amount (kg) | Protein (kg) | Amount (kg) | Protein (kg) | Amount (kg) | Protein (kg) | Amount (kg) | Protein (kg) |
|---|---|---|---|---|---|---|---|---|
| 0–4 | 1.63 ± 0.10 | 0.65 ± 0.04 | 2.33 ± 0.15 | 0.93 ± 0.06 | 5.32 ± 0.94 | 0.27 ± 0.02 | 1.95 ± 0.04 | 0.16 ± 0.00 |
| 4–8 | 10.40 ± 1.16 | 4.16 ± 0.46 | 15.08 ± 0.79 | 6.03 ± 0.32 | 1.48 ± 0.79 | 0.08 ± 0.02 | N/A | N/A |
| 8–12 | 27.64 ± 1.83 | 11.05 ± 0.73 | 29.39 ± 2.23 | 11.75 ± 0.89 | N/A | N/A | N/A | N/A |
| 12–15 | 21.37 ± 1.50 | 8.55 ± 0.60 | 25.23 ± 0.88 | 10.09 ± 0.35 | N/A | N/A | N/A | N/A |

Values shown are the means ± SD for three replicate ponds.

Protein amounts were calculated from the protein component ratio in each feed material (Table 2).

N/A: Not applicable because all benthic organisms were consumed.

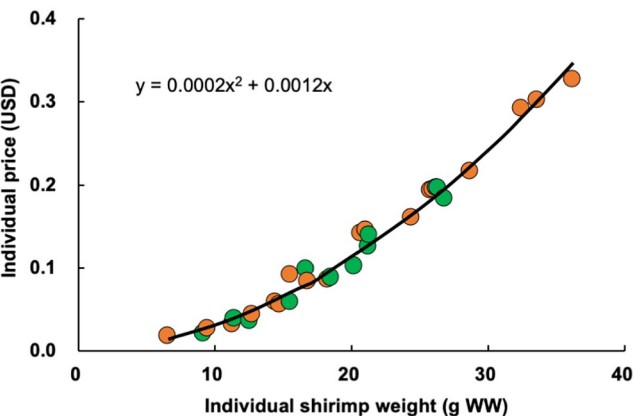

**Fig 5. Price curve for giant tiger prawn (*Penaeus monodon*) at a shrimp broker company in Bangkok, Thailand, in December 2016.** Green circle, control; orange circle, treatment. Each point was calculated based on the shrimp size category and its associated price. WW, wet weight.

**Table 4. Water quality parameters during the experimental period under the control (fed only artificial feed) and experimental treatment (fed artificial feed and benthos) conditions.**

| | Control (*n* = 3) | Experimental treatment (*n* = 3) |
|---|---|---|
| Water temperature (˚C) | 29.6 (26.9–31.2) [a] | 29.5 (26.8–31.1) [a] |
| pH | 8.0 (7.1–8.8) [a] | 7.9 (7.0–8.4) [a] |
| Salinity (PSU) | 23.7 (18.0–32.0) [a] | 23.7 (18.0–32.0) [a] |
| Alkalinity (mg $L^{-1}$) | 104.8 (58.0–120.0) [a] | 106.5 (75.0–118.0) [a] |
| Total ammonia-N (mg $L^{-1}$) | 1.7 (0.58–5.11) [a] | 1.3 (0.54–4.87) [a] |
| Nitrite (mg $L^{-1}$) | 2.2 (0.002–8.159) [a] | 2.3 (0.002–6.625) [a] |
| Nitrate (mg $L^{-1}$) | 1.5 (0.01–13.36) [a] | 1.6 (0.01–9.90) [a] |
| Phosphate (mg $L^{-1}$) | 0.8 (0.02–6.05) [a] | 0.6 (0.01–3.75) [a] |

Values shown are the means (ranges) for three replicate ponds.

Different superscript letters within the same row indicate a significant difference between treatments (Student's t-test, $p < 0.05$).

**Table 5. Comparison of results for the different benthic organisms as supplementary feeds for tropical Penaeidae.**

| Penaeidae species | Giant tiger prawn | | | | White-leg shrimp | | |
|---|---|---|---|---|---|---|---|
| Benthos species | Chs, Sts | Chs | Agt | Agt | Ulc | Uls | Cls |
| **Experimental design** | | | | | | | |
| Culture water volume (L) | 97,200 | 70 | 100 | | 2,000 | 100 | 100 |
| Shrimp density (individuals m$^{-3}$) | 27.7 | 71.4 | 100 | 150 | 25 | 200 | 200 |
| Amount of benthos prepared (g m$^{-3}$) | 70 (Chs), 20 (Sts) | Satiation | 1,000 | 1,000 | 750 | 1,000 | 1,000 |
| Culture duration (days) | 105 | 70 | 60 | 90 | 45 | 72 | 72 |
| Initial shrimp weight (g WW) | 0.002 [NS] | 0.39 [NS] | 0.46–0.51 [NS] | 1.79 [NS] | 3.5 [NS] | 0.036 [NS] | 0.036 [NS] |
| **Ratio of best value in treatment to control** | | | | | | | |
| Final shrimp weight (%) | 113.4 [S] | 156.6 [S] | 129.6 [S] | 115.1 [S] | 130.1 [S] | 142.0 [S] | 140.0 [S] |
| Final SGR (%) | 103.3 [S] | 115.7 [S] | 108.9 [S] | 108.0 [S] | 187.8 [S] | 107.7 [S] | 107.4 [S] |
| Final FCR (%) | 88.7 [S] | 61.1 [S] | 37.0 [S] | 61.7 [S] | 62.1 [S] | 34.8 [S] | 36.5 [S] |
| Survival rate (%) | 117.5 [S] | | 103.8 [S] | 121.3 [NS] | 106.7 [NS] | 107.6 [S] | 105.7 [S] |
| Reference | Present study | [16] | [17] | [15] | [11] | [14] | [14] |

Chs, *Chaetomorpha* sp.; Sts, *Stenothyra* sp.; Agt, *Agarophyton tenuistipitata* (formerly *Gracilaria tenuistipitata*); Ulc, *Ulva clathrata*; Uls, *Ulva* (formerly *Enteromorpha*) sp.; Cls, Cladophoraceae sp.; WW, wet weight; SGR, specific growth rate; FCR, feed conversion ratio; NS, no significant difference between the treatment and control; S, significant difference between the treatment and control.

the present study, shrimp growth was significantly enhanced in the experimental group, albeit at slightly lower rates than those of the previous studies.

Past studies on tropical Penaeidae fed benthic species as supplementary feeds have mainly been undertaken on a small scale in laboratories or indoor facilities, and they have provided relatively higher ratios of benthos (750 g m$^{-3}$ to satiation) compared with the number of shrimp and/or water volumes (Table 5). However, it would be difficult to preserve or prepare such a high ratio of benthos organisms during the entire culture period in real intensive shrimp ponds. For example, 500 g WW m$^{-3}$ of benthos is equivalent to 5 tons WW of benthos in an intensive shrimp pond (with an area of 1 ha and a water depth of 1.0 m), the collection and maintenance of which would require considerable effort and financial costs. Therefore, in the present study, the amount of benthos supplied was limited to the minimum amount necessary, with initial inputs of approximately 70 g m$^{-3}$ *Chaetomorpha* sp. and 20 g m$^{-3}$ *Stenothyra* sp. Therefore, the lower growth promotion in the present study likely resulted from the lower ratio of benthos provided.

Penaeidae growth follows a logistic curve [25–28]; therefore, higher production is expected with higher growth at earlier culture stages because of the exponential phase of the logistic curve. Here, the monthly individual shrimp weight and monthly SGR were significantly higher in the experimental group than in the control group during the first 4 weeks of culture (Figs 2 and 3). Moreover, the protein contributions of both benthic organisms provided in the experimental treatment ponds were higher in the first 4 weeks than after this period (Table 3). These results suggest that *Chaetomorpha* sp. and *Stenothyra* sp. effectively promoted shrimp growth as supplementary live feeds during the first 4 weeks in the early stages of growth, leading to the higher final production.

Compared with when they were given artificial feeds only, Penaeidae exhibited enhanced survival rates when offered several types of benthic species, with improvements of 104%–121% for giant tiger prawn and 106%–108% for white-leg shrimp (Table 5). Similarly, a significantly higher survival rate was observed in the experimental treatment than in the control treatment in the present study (Table 5). The reasons for this improvement in survival rates are currently unclear and require further study.

The global productivity of giant tiger prawn reportedly ranges from 0.4–1.5 kg m$^{-2}$ year$^{-1}$ in intensive earthen culture ponds [29]. Concrete ponds are generally associated with lower productivity than earthen intensive culture ponds. However, the productivity of giant tiger prawn in the experimental concrete ponds used in this study was approximately 0.5 kg m$^{-2}$ following a 4-month culture period. Assuming a culture frequency of two times per year including the preparation period, giant tiger shrimp productivity with this system is expected to produce approximately 1.0 kg m$^{-2}$ year$^{-1}$ of giant tiger prawn. Therefore, the productivity achieved in this study was not inferior to real earthen intensive shrimp ponds, indicating that the data obtained will be applicable to these systems.

The quantities of benthic organisms provided in the experimental treatment ponds were designated to minimize the possibility of all shrimp being consumed within 1–2 month(s), to lessen the need for additional labor. The addition of more benthic organisms would have likely led to further improvements in growth and survival. However, the efforts that would be required to collect additional organisms would not be feasible in a real facility owing to the added time, labor, and financial costs it would bring to small-scale shrimp farmers. Therefore, simple techniques to increase the number of benthic organisms are required, such as the self-propagation of benthos in shrimp ponds.

## Feed consumption and efficiency

A study by [11] mentioned that aFCR refers solely to artificial feed as a food source and excludes the contribution of other food materials, such as seaweeds. This expression is also useful for visualizing feed saving and for making a practical estimation of the feed cost. Previous studies have reported that the aFCRs of Penaeidae are significantly lower when benthic organisms are fed as supplementary feeds than when artificial feed only is offered, with values in "supplementary feed" groups ranging from 37% to 62% of those in the "artificial feed only" groups for giant tiger prawn [15–17] and from 35% to 62% for white-leg shrimp [11, 14] (Table 5). In the present study, the aFCR was significantly lower (89%) in the supplementary feed group than the artificial feed only group (Tables 1 and 5). Although the aFCR obtained here was slightly higher (i.e., the feed efficiency was lower) than the values reported in previous studies (because of the smaller ratio of benthic organisms provided in this study), it is clear that the benthic organisms were efficiently utilized for shrimp production.

## Cost and profitability

When considering the application of innovative techniques to real intensive shrimp ponds, it is necessary to determine their profitability such that the benefits may be predicted and extrapolated. However, very few studies on benthic organisms as supplementary feeds for tropical Penaeidae have been scaled up, and shrimp profitability has not been discussed [11–15, 17, 18]. Past studies evaluated only shrimp growth or aFCR using small water containers. This is because it is difficult to obtain the number of synchronously operating earthen shrimp ponds required for statistical replication. In the present study, culturing PL to a marketable size using the outdoor concrete experimental ponds allowed for a balanced assessment of experimental shrimp sales data and statistical analysis.

The individual shrimp prices obtained in this study followed an exponential curve (Fig 5). Larger numbers of bigger shrimp were observed in the experimental treatment than in the control (Fig 4). Higher sales were possible for the experimental group owing to the higher mean weight of individual shrimp, total number of shrimp, and total shrimp production (Table 1). Although the experimental group was also associated with higher feed costs and

miscellaneous costs for benthos collection, the shrimp sales largely exceeded these costs, resulting in higher profitability.

It has previously been reported that several benthic species of seaweeds and snails enhance not only shrimp growth [16, 30] and feed efficiency [31], but also shrimp immunity, resistance to thermal stress, and the viral challenges of disease [32–34], and color [11, 14, 15, 17] when benthic species were fed as supplementary feeds. Therefore, although only the productivity, feed efficiency, and profitability of giant tiger prawn were considered here, *Chaetomorpha* sp. and *Stenothyra* sp. may also confer these additional benefits. More detailed studies are needed.

Finally, additional effort, laborers, and costs (e.g., associated with collection and propagation in ponds) for benthic organisms and collection are usually needed to operate a system with benthic organisms as supplementary feeds. However, this study demonstrated that these can be restricted to an acceptable range, making it technically feasible to apply this technique to intensive shrimp culture ponds. In the near future, the authors will attempt to expand this experimental culture operation on a commercial scale. To reduce benthos collection efforts and costs for farmers, as well as the negative impacts of shrimp culture on natural resources, future studies will examine simple techniques to increase the amounts and/or lifespans of these benthos (e.g., self-propagation in earthen ponds).

## Conclusions

To accelerate the implementation of an innovative system in intensive shrimp aquaculture ponds, the productivity, feed efficiency, and profitability of giant tiger prawn were evaluated for a system in which a filamentous green alga (*Chaetomorpha* sp.) and a microsnail (*Stenothyra* sp.) are propagated together during an early culture stage and freely consumed as supplementary feeds in outdoor concrete ponds. Approximately 44 kg of giant tiger prawn was produced in total from an initial weight of 5.4 g of PL (0.002 g WW individual$^{-1}$ and 2,700 individuals) when fed *Chaetomorpha* sp. (initial wet weight of 6.81 kg, which is 8.37% to total feed consumption; estimated protein weight of 0.35 kg) and *Stenothyra* sp. (initial wet weight of 1.96 kg, which is 2.43% to total feed consumption; estimated protein weight of 0.16 kg) as supplementary feeds. The productivity, feed efficiency, and profitability for the group fed benthic organisms were approximately 133%, 113%, and 146%, respectively, of the values recorded for the artificial feed only group. Moreover, the effort and costs associated with the provision of benthic organisms could be restricted to an acceptable range. In conclusion, the application of *Chaetomorpha* sp. and *Stenothyra* sp. as live feeds at an early stage of intensive aquaculture will improve initial shrimp growth performance, which will subsequently lead to the promotion of shrimp productivity, feed efficiency, and profitability. This innovative technique may enhance the efficiency of intensive aquaculture for the giant tiger prawn, and it is technically feasible using earthen shrimp culture ponds.

## Acknowledgments

We thank the NRCT for advising us to conduct this study in Thailand. We are also grateful to Associate Professor Dr. Paveena Taveekijakarn, the head of the Fisheries Science Program of KMITL, for their helpful advice. Our appreciation also goes to Associate Professor Dr. Yomla Rungtawan of KMITL for supporting the experiment. We thank Dr. Tomoyuki Okutsu and Dr. Ryogen Nambu of JIRCAS for providing helpful suggestions. We thank Dr. Yukiyo Yamamoto and Dr. Tetsuo Fujii of JIRCAS for their encouraging comments. We appreciate Prof. Emeritus Dr. Masao Ohno of Kochi University for his constructive advice. Finally, we thank Natalie Kim, PhD of Edanz Group (https://en-author-services.edanzgroup.com/ac) for editing a draft of this manuscript and helping to develop the abstract.

## Author Contributions

**Conceptualization:** Isao Tsutsui, Kaoru Hamano.

**Data curation:** Isao Tsutsui, Piyarat Pinphoo.

**Formal analysis:** Isao Tsutsui, Piyarat Pinphoo, Worachet Thuamsuwan, Jaruwan Songphatkaew.

**Funding acquisition:** Osamu Abe.

**Investigation:** Isao Tsutsui, Dusit Aue-umneoy, Piyarat Pinphoo, Worachet Thuamsuwan, Kittipong Janeauksorn, Grissada Meethong, Patcharanut Keattanaworada, Jaruwan Songphatkaew.

**Methodology:** Isao Tsutsui, Dusit Aue-umneoy, Piyarat Pinphoo.

**Project administration:** Isao Tsutsui, Osamu Abe.

**Resources:** Dusit Aue-umneoy, Piyarat Pinphoo.

**Supervision:** Monthon Ganmanee, Osamu Abe, Kaoru Hamano.

**Visualization:** Dusit Aue-umneoy.

**Writing – original draft:** Isao Tsutsui, Dusit Aue-umneoy.

**Writing – review & editing:** Monthon Ganmanee, Osamu Abe, Kaoru Hamano.

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
