## [Decision Letter · Decision Letter 0]

28 Sep 2020

PONE-D-20-21866

Use of a filamentous green alga (Chaetomorpha sp.) and microsnail (Stenothyra sp.) as feed at an early stage of intensive aquaculture promotes growth performance, artificial feed efficiency, and profitability of giant tiger prawn (Penaeus monodon)

PLOS ONE

Dear Dr. Tsutsui,

Thank you for submitting your manuscript to PLOS ONE. After careful consideration, we feel that it has merit but does not fully meet PLOS ONE’s publication criteria as it currently stands. Therefore, we invite you to submit a revised version of the manuscript that addresses the points raised during the review process.

We look forward to receiving your revised manuscript.

Kind regards,

Mahmoud A.O. Dawood, PhD

Academic Editor

PLOS ONE

Journal Requirements:

2.Thank you for stating the following financial disclosure:

 [The funders had no role in study design, data collection and analysis, decision to publish, or preparation of the manuscript.].

Reviewers' comments:

Reviewer's Responses to Questions

**Comments to the Author**

1. Is the manuscript technically sound, and do the data support the conclusions?

Reviewer #1: Partly

Reviewer #2: Partly

Reviewer #3: Yes

Reviewer #4: Yes

2. Has the statistical analysis been performed appropriately and rigorously? 

Reviewer #1: Yes

Reviewer #2: No

Reviewer #3: No

Reviewer #4: Yes

3. Have the authors made all data underlying the findings in their manuscript fully available?

Reviewer #1: Yes

Reviewer #2: Yes

Reviewer #3: Yes

Reviewer #4: Yes

4. Is the manuscript presented in an intelligible fashion and written in standard English?

Reviewer #1: Yes

Reviewer #2: Yes

Reviewer #3: Yes

Reviewer #4: No

5. Review Comments to the Author

Reviewer #1: Author used Chaetomorpha sp. and Stenothyra sp. as feed at an early stage of intensive aquaculture of giant tiger prawn. However, there are still some problems should be improved, especially the experimental method and calculation. The following questions should be answered before further submission.

Abstract: It is hard to get to know the main content of this manuscript form the abstract. Author should provide more effective information in concise sentences.

Line 27-31: Confused. Please rewrite it.

Line 32-37: Rewrite. More results should be exhibited in abstract.

Introduction:

Line 69: References are needed here.

Line 75-82: Redundant content. Why you choose Chaetomorpha sp. and Stenothyra sp.?

Line 84-85: Why culture both Chaetomorpha sp. and Stenothyra sp. as supplementary live feeds? Why use them as feed at an early stage of intensive aquaculture?

Materials and methods:

Line 120: Why choose these amounts?

Line 121-123: How to remove other benthic organisms?

Line 123-124: How about the daily management?

Line 134: Only 20 shrimps? Compared with the initial number of shrimps, the measured quantity is quite small, whether it leads to a big error?

Discussion：

Line 280-283: Confused. Please explain these sentences.

Table 1: How do you calculate the apparent benthos consumption?

Table 3: There are no Chaetomorpha sp. Stenothyra sp. provided after 8 weeks. Why the higher feed consumption was found in experimental treatment?

Table 4: 0 may not possible. Please explain.

Table 5: What is the amount of benthos? What is the amount of benthos? Line 120: 6.81 kg and 1.96 kg. Survival rate>100%???

Reviewer #2: Apparently the experimental diet stimulated growth; the performance of P. monodon appears to have improved in the 3 groups given supplemental feeds as PLs as compared with controls. This comparison is limited statistically by the small number of groups (N = 3) and apparent variation in food consumption and survival. For these reasons I suggest that this be correctly described as as a pilot or preliminary observation and not presented as a definitive analysis. When an experimental treatment appears to hold the promise of increasing commercial production and improved profitability for a valuable crop, is is reasonable to test it on a scale that is adequate for the use of rigorous statistical analysis.

Was any cost or value ascribed to the green algae and snails that were used as supplemental feeds in this study? It is not clear to me if these organisms would be available to scale up for culture on a commercial scale. The experimental group ate more of the prepared commercial feed, so at least one mechanism of action was an increased appetite in response to the early supplemental feeds.

The discussion of IMTA on page 18 is superfluous; as the authors state, this study is not an example of integrated aquaculture, it is the addition of live feeds.

The abstract refers to the culture of “an early stage” of P. monodon; it would be preferable to identify them specifically as P. monodon postlarvae.

Line 217 Penaeus monodon should be italicized for correct scientific name format.

The different survival rates with such a low number of samples (3 control and 3 experimental ponds) is troubling – as the authors mentioned more than once, this experimental design does not allow for strong and definitive statistics. With such a small number of samples, random variation or tank effects can not be ruled out.

Reviewer #3: Table 1. check your value.

If 16.0 ± 6.1 is correct, it might be strongly possible that the statistics also are subject to change. 6.1 is high so I didn't see a statistical difference between 16.0 and 18.2.

Figures.

The figures are low resolution, which makes their observation difficult. I suggest increasing the resolution.

Reviewer #4: The authors are encourage to address the following issues:

1- The article language needs edition.

2- Abstract needs restructuring. It contains much background data and less methodological data and results. Revise it.

3- It is not clear if the added live food reproduced in the culture tank during 15 weeks of experiment. Clarify it.

4- In table 1, "total shrimp production": Unit is confusing. kg shrimp per tank? per squared meter?... Moreover, add feed intake as percentage of body weight.

5-

6. PLOS authors have the option to publish the peer review history of their article (what does this mean?). If published, this will include your full peer review and any attached files.

Reviewer #1: No

Reviewer #2: No

Reviewer #3: **Yes: **Sevdan YILMAZ

Reviewer #4: No

---

## [Author Response · Author response to Decision Letter 0]

9 Nov 2020

Dear Dr. Dawood

Academic editor,

PLOS ONE

Thank you very much Dr. Dawood and each of the reviewers for valuable insights. We are delighted that you think our work will spark discussion in our field. We are grateful for the time and energy you have expended on our behalf.

I attached a file named "Response to Reviewers" on this submission web-site, as our responses and answers to each of the reviewers’ questions and comments.

Thank you again for insightful comments and for providing us with the chance to revise our manuscript.

Sincerely yours;

Isao Tsutsui

---

## [Decision Letter · Decision Letter 1]

23 Nov 2020

PONE-D-20-21866R1

Use of a filamentous green alga (Chaetomorpha sp.) and microsnail (Stenothyra sp.) as feed at an early stage of intensive aquaculture promotes growth performance, artificial feed efficiency, and profitability of giant tiger prawn (Penaeus monodon)

PLOS ONE

Dear Dr. Tsutsui,

Thank you for submitting your manuscript to PLOS ONE. After careful consideration, we feel that it has merit but does not fully meet PLOS ONE’s publication criteria as it currently stands. Therefore, we invite you to submit a revised version of the manuscript that addresses the points raised during the review process.

We look forward to receiving your revised manuscript.

Kind regards,

Mahmoud A.O. Dawood, PhD

Academic Editor

PLOS ONE

Additional Editor Comments (if provided):

Authors are recommended to address the comments of reviewer 4 that has been raised in the first round of review as following:

1- The article language needs edition.

2- Abstract needs restructuring. It contains much background data and less methodological data and results. Revise it.

3- It is not clear if the added live food reproduced in the culture tank during 15 weeks of experiment. Clarify it.

4- In table 1, "total shrimp production": Unit is confusing. kg shrimp per tank? per squared meter?... Moreover, add feed intake as percentage of body weight.

Also, do you have response to the concern raised by reviewer 2?

Reviewers' comments:

Reviewer's Responses to Questions

**Comments to the Author**

1. If the authors have adequately addressed your comments raised in a previous round of review and you feel that this manuscript is now acceptable for publication, you may indicate that here to bypass the “Comments to the Author” section, enter your conflict of interest statement in the “Confidential to Editor” section, and submit your "Accept" recommendation.

Reviewer #2: All comments have been addressed

Reviewer #3: All comments have been addressed

Reviewer #4: (No Response)

2. Is the manuscript technically sound, and do the data support the conclusions?

Reviewer #2: Yes

Reviewer #3: Yes

Reviewer #4: Partly

3. Has the statistical analysis been performed appropriately and rigorously? 

Reviewer #2: Yes

Reviewer #3: Yes

Reviewer #4: I Don't Know

4. Have the authors made all data underlying the findings in their manuscript fully available?

Reviewer #2: Yes

Reviewer #3: Yes

Reviewer #4: Yes

5. Is the manuscript presented in an intelligible fashion and written in standard English?

Reviewer #2: Yes

Reviewer #3: Yes

Reviewer #4: No

6. Review Comments to the Author

Reviewer #2: The authors have argued that this study with N=3 samples is statistically valid because of the mangnitude of differences related to treatment vs control. They have made a good case statistically, and the contribution is in an acceptable form, and other comments in my initial review have been addressed satisfactorily. I am inherently uncomfortable with such small sample sizes and have done farm-scale culture studies with an N of nine, on nine commercial farms, which in my view is preferable and more convincing than making a case with a study on such a small scale.

Reviewer #3: The manuscript entitled " Use of a filamentous green alga (Chaetomorpha sp.) and microsnail (Stenothyra sp.) as feed at an early stage of intensive aquaculture promotes growth performance, artificial feed efficiency, and profitability of giant tiger prawn (Penaeus monodon) " is a well written work; the topic is very interesting for journal reader. I have reviewed revised manuscript. Many parts of revision seemed to be adequate.

Therefore, I judged this manuscript would be acceptable.

Reviewer #4: (No Response)

7. PLOS authors have the option to publish the peer review history of their article (what does this mean?). If published, this will include your full peer review and any attached files.

Reviewer #2: **Yes: **Christopher L. Brown

Reviewer #3: No

Reviewer #4: No

---

## [Author Response · Author response to Decision Letter 1]

4 Dec 2020

Mahmoud A.O. Dawood, PhD

Academic Editor

PLOS ONE

4 December 2020

Submission. No.: PONE-D-20-21866

Title: Use of a filamentous green alga (Chaetomorpha sp.) and microsnail (Stenothyra sp.) as feed at an early stage of intensive aquaculture promotes growth performance, artificial feed efficiency, and profitability of giant tiger prawn (Penaeus monodon)

Dear Dr. Dawood

Thank you for your further critical comments on our manuscript, and for those of the reviewers. We are delighted to hear that you think our work will spark debate in our field, and are grateful for the time and energy you have expended on our behalf. Our responses to the points made by the Academic Editor and by each of the reviewers in their second review are shown in a file named “Response to Reviewers“.

Thank you very much again. I look forward to hearing from you at your earliest convenience.

Kind regards,

Isao Tsutsui

Fisheries Division,

Japan International Research Center for Agricultural Sciences (JIRCAS)

1-1 Ohwashi, Tsukuba, Ibaraki 305-8686, Japan

Phone: +81-(0)29- 838-6608

Fax: +81-(0)29- 838-6655

E-mail: cong@affrc.go.jp

---

## [Decision Letter · Decision Letter 2]

14 Dec 2020

Use of a filamentous green alga (Chaetomorpha sp.) and microsnail (Stenothyra sp.) as feed at an early stage of intensive aquaculture promotes growth performance, artificial feed efficiency, and profitability of giant tiger prawn (Penaeus monodon)

PONE-D-20-21866R2

Dear Dr. Tsutsui,

We’re pleased to inform you that your manuscript has been judged scientifically suitable for publication and will be formally accepted for publication once it meets all outstanding technical requirements.

Kind regards,

Mahmoud A.O. Dawood, PhD

Academic Editor

PLOS ONE

Additional Editor Comments (optional):

Reviewers' comments:

Reviewer's Responses to Questions

**Comments to the Author**

1. If the authors have adequately addressed your comments raised in a previous round of review and you feel that this manuscript is now acceptable for publication, you may indicate that here to bypass the “Comments to the Author” section, enter your conflict of interest statement in the “Confidential to Editor” section, and submit your "Accept" recommendation.

Reviewer #2: All comments have been addressed

Reviewer #4: (No Response)

2. Is the manuscript technically sound, and do the data support the conclusions?

Reviewer #2: Yes

Reviewer #4: Yes

3. Has the statistical analysis been performed appropriately and rigorously? 

Reviewer #2: Yes

Reviewer #4: Yes

4. Have the authors made all data underlying the findings in their manuscript fully available?

Reviewer #2: Yes

Reviewer #4: Yes

5. Is the manuscript presented in an intelligible fashion and written in standard English?

Reviewer #2: Yes

Reviewer #4: Yes

6. Review Comments to the Author

Reviewer #2: This is my third review of this manuscript, and I was already satisfied with the revisions included in the second one. Again, the number of replicate samples is very small but the statististical significance appears to be satisfactory.

Reviewer #4: The authors revised the manuscript and it is acceptable. I recommend it for pulication///////////////

7. PLOS authors have the option to publish the peer review history of their article (what does this mean?). If published, this will include your full peer review and any attached files.

Reviewer #2: **Yes: **Chrisotpher L. Brown

Reviewer #4: No

---

## [Editor Report · Acceptance letter]

16 Dec 2020

PONE-D-20-21866R2 

Use of a filamentous green alga (*Chaetomorpha sp.*) and microsnail (*Stenothyra sp.*) as feed at an early stage... 

Dear Dr. Tsutsui:

I'm pleased to inform you that your manuscript has been deemed suitable for publication in PLOS ONE. Congratulations! Your manuscript is now with our production department. 

Kind regards, 

on behalf of

Dr. Mahmoud A.O. Dawood 

Academic Editor

PLOS ONE